# Food insecurity status and determinants among Urban Productive Safety Net Program beneficiary households in Addis Ababa, Ethiopia

Atimen Derso[1][☯], Hailemichael Bizuneh[2], Awoke Keleb[1], Ayechew Ademas[1], Metadel Adane[1][☯]*

1 Department of Environmental Health, College of Medicine and Health Sciences, Wollo University, Dessie, Ethiopia, 2 Department of Public Health, Saint Paul's Hospital Millennium Medical College, Addis Ababa, Ethiopia

☯ These authors contributed equally to this work.
* metadel.adane2@gmail.com

**Data Availability Statement:** All relevant data are within the manuscript and its Supporting Information files.

## Abstract

### Background

Measuring household food insecurity in specific geographic areas provides vital information that enables appropriate and effective intervention measures to be taken. To that end, this study aimed to assess the prevalence of food insecurity and associated factors among Urban Productive Safety Net Program (UPSNP) beneficiary households in Addis Ababa, Ethiopia's capital city.

### Methods

A community-based cross-sectional study was conducted among 624 UPSNP beneficiary households in nine districts of Addis Ababa from June to July 2019. A multi-stage sampling method was used; study participants were selected using a simple random sampling technique after establishing the proportionally allocated sample size for 9 districts. Data were collected by trained personnel using a pretested, structured questionnaire. The outcome variable was food insecurity as measured by Household Food Insecurity Access Scale (HFIAS), a tool developed by the Food and Nutrition Technical Assistance Scale (FANTA) and validated for developing countries, including Ethiopia. A binary (crude odds ratio [COR]) and multivariable (adjusted odds ratio [AOR]) logistic regression analysis were employed at 95% CI (confidence interval). From the bivariate analysis, factors having a $p$-value<0.25 were included in the multivariable analysis. From the multivariable analysis, any variable at $p$-value < 0.05 at 95% CI was declared significantly associated with household food insecurity. Model fitness was also checked using the Hosmer-Lemeshow test with $p$-value>0.05.

### Results

The prevalence of household food insecurity was 77.1% [95%CI:73.8–80.7] during the month prior to the survey. Illiteracy of household head [AOR: 2.56; 95%CI:1.08–6.07], family

**Funding:** Funding for this study was provided by Saint Paul's Hospital Millennium Medical College. The funders had no role in study design, data collection and analysis, decision to publish, or preparation of the manuscript.

**Competing interests:** The authors have declared that no competing interests exist.

**Abbreviations:** AOR, Adjusted odds ratio; COR, Crude odds ratio; FANTA, Food and Nutrition Technical scale; HFIAS, Household food insecurity access scale; UPSNP, Urban productive safety net program; USD, United States Dollar; CI, Confidence Interval; EDHS, Ethiopian Demographic and Health Survey; UNICEF, United Nations International Children's Emergency.

size of 4 or more [AOR: 1.87, 95%CI:1.08–3.23], high dependency ratio [AOR: 3.95; 95% CI:1.31–11.90], household lack of access to credit [AOR:2.85; 95%CI:1.25–6.49], low household income [AOR: 4.72; 95%CI:2.32–9.60] and medium household income [AOR: 9.78; 95%CI:4.29–22.35] were significantly associated with household food insecurity.

## Conclusion

We found that three in four of Addis Ababa's UPSNP beneficiary households were food-insecure. Implementation of measures to improve household income, minimize the dependency ratio of households, and arrange access to credit services are paramount ways to tackle food insecurity problems in Addis Ababa.

## Introduction

Household food insecurity exists when a family does not have adequate physical, social and economic access to sufficient, safe and nutritious food to meet the dietary needs and food preferences of its members for an active and healthy life [1, 2]. It is one of the underlying causes of all forms of malnutrition, including inadequate quantity, poor quality, and continuity of diet [3, 4] that persists as a major challenge around the world.

Lack of access to sufficient and nutritious food is a significant challenge to achieving international nutritional targets for children. For instance, according to the 2018 Global Nutrition Report, only half (51%) of children aged 6 to 23 months around the world get the recommended minimum number of daily meals, and only one in five children (16%) eat a minimally acceptable diet [5]. From a 2019 report by the United Nations International Children's Emergency Fund (UNICEF) on the state of the world's children, by 2018 the proportion of children between 6 and 23 months of age eating a diverse diet is only one in three. However, for the world's poorest children, the ratio falls to only one in five [6].

The prevalence of poverty and the urban population in Africa are growing rapidly [7] Both of these factors are increasing the challenge of meeting food security needs of the urban population. Members of this poor urban population experience under-nutrition, micronutrient deficiencies, and being overweight and obese at the same time, increasing the double burden of malnutrition [8]. Currently, one in three undernourished children live in an urban setting, a proportion that will increase over the next decade in response to globalization, migration, population growth, income inequality and climate change across low- and middle-income countries [9–11].

In Africa, food insecurity is a major public health problem that appears to be increasing in almost all regions, causing 52.5% of the population to be exposed to moderate or severe food insecurity. In 2018, among populations of Sub-Sahara and Eastern Africa, the magnitude of food insecurity was 57.7% and 62.7% respectively.

In eastern Africa, as a consequence of food insecurity, nearly one-third of the population (30.8%) was undernourished due to a lack of sufficient calories [3].

Food insecurity is manifested in the reduction of food consumption and a lack of access to food variety and diversity [12, 13], affecting per capita food consumption. Inadequate access to food causes inequality of food distribution among family members, especially a reduction to infants and children [14], as well as anxiety and stress among women in the household [15]. Ultimately, a lack of access to food affects the physical and mental development of an individual and the economic productivity of a population and a country [3, 16].

According to a 2018 global report on food crises, the severity of food insecurity in Ethiopia increased proportionally between 2016 and 2017 despite a decrease in absolute numbers;

around 8.5 million people lack food security as a result of population displacement, political instability, armed clashes, persistent drought and an increase in food prices [17]. Food insecurity has an indirect significant contribution to under-nutrition that ultimately affects the economic growth of a country. According to the African Union (AU) 2014 report on the cost of hunger in Africa, under-nutrition in Ethiopia costs 16.5% of gross domestic product (GDP) and causes 28% of under-five child mortality, 15.8% of students repeating a grade, and nearly 13 billion Ethiopian birr lost in productivity in manual activities [18].

From the 2016 Ethiopian Demographic Health Survey (EDHS) report, only 45% of children aged 6 to 23 months had gotten the recommended minimum number of meals, and only 7% of them had eaten a minimally acceptable diet in the previous five years [19]. Great efforts have been made by Ethiopian social protection programs. Evidence shows that a rural safety net program and cash and food transfers to households has improved food security and dietary diversity of low-income rural households; the program has now been expanded to Ethiopia's urban settings [20].

The UPSNP is a pilot project launched in 2017 by the Ethiopian government and aid organizations to help urban poor food insecure households to improve nutritional status and address the underlying causes of food insecurity by providing them with food and cash [16]. Addis Ababa is one of the pilot project areas. However, the status of food insecurity among urban beneficiary households has not been assessed since starting the program. This study was developed to address this gap in knowledge which will help programmers to evaluate and revisit the approaches of the program in an urban setting.

Therefore, this study aims to gather area-specific evidence to assess the status of food insecurity and associated factors among UPSNP beneficiary households in Ethiopia's capital city, Addis Ababa. A key goal of this study is to contribute to the scaling up of the UPSNP, thereby helping to change the food security status of Addis Ababa's urban poor.

## Methods and materials

### Study design, period and study area

A community-based cross-sectional study was conducted from June to July 2019 in Addis Ababa, Ethiopia. This study covered 9 districts (called *woredas* in Amharic, the local language) found in three sub-cities of Addis Ababa (Arada, Lideta and Yeka). However, in all, the UPSNP was implemented in 90 selected districts (of 116 existing districts) across all 10 sub-cities. About 78,543 households have been participating in UPSNP since 2017 [21]. According to the projection of the 2007 Ethiopian population census, the total population of the city in 2017 was 4,567,857 with an estimated 913,572 households [22].

The national nutrition policy of Ethiopia addresses food insecurity through preventive approaches via nutrition-sensitive interventions that address underlying causes such as poverty [23]. The government of Ethiopia, the World Food Program (WFP) and development partners have worked together to improve food security, and to stabilize assets covering 10 million Ethiopian people through the UPSNP [21].

The UPSNP targets urban poor households (those earning below 2 United States Dollars [USD] per day) with nutritionally vulnerable members in the identified pilot cities including Addis Ababa and some regional towns with interventions that improve their living and food security levels [21].

### Source population, inclusion and exclusion criteria

All UPSNP beneficiary households in Addis Ababa were the source population of the study. The study population was randomly selected UPSNP beneficiary households in the

systematically selected districts. From the study population, household respondents who were unable to communicate for various reasons such as illness were excluded. Registered and active UPSNP beneficiary households in the selected districts at the time of data collection were included.

## Sample size determination and sampling procedure

To estimate sample size, we considered the assumptions of sample estimator, previous prevalence (proportion of food insecurity), sample variability and limit of uncertainty (95% confidence interval). Here, we assumed the UPSNP beneficiary population is approximately normally distributed then standard error of population value 'p' is $SE_p = \sqrt{P(1-P)/n}$, then P (population value) = p(sample value) ± z SE$_P$

± $Z_{\alpha/2}$ * SE$_P$ = variability of population value (W),

$$W = Z_{\alpha/2} * SE_P = z\sqrt{P(1-P)/n}$$

Finally, the sample size was determined using a single proportion formula:

$$n = \frac{[(z^{\alpha}/_2)2 * (p(1-p)]]}{w^2}$$

with the assumption of the prevalence (p) of household food insecurity at 58.16%, taken from a previous study in Addis Ababa, Ethiopia [24], with a 95% confidence interval (CI) ($Z_{a/2}$ = 1.96) and (w) marginal error (5%). A 1.5 design effect was also considered, and then 10% contingency for non-response rate was included. Finally, we obtained a final sample size of 624 households.

We used multi-stage sampling in two stages. During the first stage, out of Addis Ababa's 10 sub-cities, 30% were selected (the 3 sub-cities Arada, Lideta and Yeka) by probability proportional to size sampling. Then, in the second stage, from the selected sub-cities, 30% of the 30 active districts implementing the program were again selected by probability proportional to size sampling, which is 9 districts; and then sample size was proportionally allocated to the 9 districts. Finally, using their payment code list already available in the respective districts sampling unit (household) was selected using systematic sampling techniques. From the selected households, the study participants were selected based on their experience of household food status according to FANTA guidelines [25].

## Outcome variable measurement

This study's outcome variable was household food insecurity status ["insecure household = 1" or "secure household = 0"] during the month previous to the survey. We used a standard tool HFIAS that was able to provide urban-level data on the program, a tool developed by FANTA and validated for developing countries, including Ethiopia. However, the indicator of the program is not able to quantify the level of food insecurity, rather it only measures the time a household lives with the program and withdraws from the program. HFIAS has three domains: anxiety and uncertainty domain, insufficient quality domain, and insufficient food intake (quantity) and its physical consequences domain. The prevalence or proportion of the three domains were evaluated based on the number of households experiencing one or more behaviours in each specific domain relative to the total number of household responding affirmatively to behaviour questions in that specific domain [25].

HFIAS has a set of nine closed-ended questions categorized as "yes" or "no", which represent occurrence; and under each "yes" category 3 option of occurrence frequency [rarely = 1,

sometimes = 2, and often = 3]. The status of household food insecurity was estimated into two categories [food insecure and food secure] using the prevalence of insecure household food access. Then, household food insecurity status was classified as mild, moderate or severe based on the frequency of the occurrence of each item in Table 1.

**A food-secure household** was declared when household score [item one = '0' or '1' and items two to nine = '0'].

**A food-insecure household** was declared when household score [item one = '2' or '3' and/or item two to nine = '0'or '1' or '2' or '3'].

**Mildly food-insecure household** [(item one = '2' or '3' or item two = '1,' '2,' or '3' or item three = '1' or item four = '1') and item five to nine = '0'].

**Moderately food-insecure household** [(item three = '2' or '3' or item four = '2' or '3' or item five = '1' or '2' or item six = '1' or '2') and item seven to nine = '0'] and

**Severely food-insecure household** score [item five = '3' or item six = '3' or item seven = '1,' '2,' or '3' or item eight = '1,' '2,' or '3' or item nine = '1,' '2,'or '3'] [25] (Table 1).

When there was a response of yes, there was an occurrence frequency measurement with three options, 'Rarely = happened once or twice', 'Sometimes = happened three to ten times' and 'Often = happened more than ten times' during the month previous to the survey [25].

## Operational definitions

**Urban Productive Safety Net Program (UPSNP).** A social protection system that is being implemented in the major urban centres of Ethiopia targeting chronic food insecure households and providing cash or food for the beneficiaries (households) either "for work", "for free" or "a combination of both" on a regular basis for a five-year period [26].

**HFIAS.** Household Food Insecurity Access Scale is a set of questions related to the experience of food access that appeared to distinguish food-secure from food-insecure households across different cultural contexts, and is used to estimate the prevalence of household food insecurity and its category [27].

**Food insecurity.** Food insecurity is a state that exists when any people do not have adequate physical, social or economic access to sufficient, safe and nutritious food that meets the dietary needs and food preferences for an active and healthy life [28].

**Recall period.** The respondent was asked an occurrence question about whether the condition had happened in the past four weeks (30 days) [25].

**Table 1. Question items 1 to 9 to measure the outcome variable food insecurity.**

| Household Food Insecurity Access Scale question item | Yes | | | No |
|---|---|---|---|---|
| Worried about not enough food | Rarely [1] | Sometimes [2] | Often [3] | No [0] |
| Unable to eat preferred food | Rarely [1] | Sometimes [2] | Often [3] | No [0] |
| Ate just a few kinds of food | Rarely [1] | Sometimes [2] | Often [3] | No [0] |
| Ate unwanted foods | Rarely [1] | Sometimes [2] | Often [3] | No [0] |
| Ate a smaller than desired amount at a meal | Rarely [1] | Sometimes [2] | Often [3] | No [0] |
| Ate fewer meals in a day than desired | Rarely [1] | Sometimes [2] | Often [3] | No [0] |
| Had no food of any kind | Rarely [1] | Sometimes [2] | Often [3] | No [0] |
| Went to sleep at night hungry | Rarely [1] | Sometimes [2] | Often [3] | No [0] |
| Went without food over a day & night | Rarely [1] | Sometimes [2] | Often [3] | No [0] |

Source:[25].

**Dependency ratio.** It was computed from the ratio of dependent members (age group less than 18 and greater than 64 years) to productive household members (age group 18–64 years) and finally sorted into three groups (tertiles) of low, medium and high.

**Household income.** It was computed into tertiles (three groups) as low (375–1500 birr), medium (1560–2000 birr) or high (2100–4500 birr) from scores of average household monthly income from any sources.

### Data collection, quality and management

A pre-tested structured questionnaire was used to collect data on socio-demographic variables and household food insecurity. The socio-demographic part was adapted from published literature and the EDHS [29]. A household food insecurity HFIAS questionnaire was adopted from the FANTA, a validated tool for food insecurity study in developing countries, including Ethiopia [25]. The questionnaire was first prepared in English and then translated to Amharic and back to English by language experts to ensure consistency. Data collectors were 4 MPH holder/Public Health Nutrition professionals; 2 supervisors were MPH holders in Public Health Nutrition.

Since Addis Ababa's UPSNP agency does not have a central database containing a list of beneficiary households and their characteristics, households were selected based on multi-stage probability proportional to size sampling design from 10 sub-cities having different sizes of beneficiary households; that design was able to weight size difference and reduce selection bias. The quality of the data was ensured through adequate training of data collectors and supervisors on the objective of the study and the overall approaches of the study. The outcome variable was measured by an HFIAS indicator that is validated, highly reliable and a culturally sound tool able to control bias and reveal the true prevalence or change in food insecurity level in this study target area.

A pre-test was conducted on 5% of the sample size in a nearby area outside of the selected study areas. Data were collected by face-to-face interviews with mothers or heads of households. The completeness of questionnaires was checked daily before data entry. The completeness of data was checked manually, and after editing and cleaning, data entry was done by using EpiData version 3.1. After checking the consistency, data were exported to Statistical Package for Social Science (SPSS) version 25.0 for analysis.

### Data analysis: Method of estimation and model equation

Prevalence of household food insecurity and level of food insecurity status (mild, moderate and severe) were computed from nine questions of the HFIAS (Table 1). To estimate the probability of the two outcome variables, the proportion of households having the characteristics of food insecurity (P) or not having the characteristics of food insecurity (1-P) computed. We also computed frequency distribution, categorization for continuous and re-categorization for categorical variables.

To determine the independent effect of predictor variables, we conducted a logistic regression model (logit transformation) written as Eqs 1 and 2.

$$\text{logit(P)} = \ln(P/1-P) = a + b_1x_1 + b_2x_2 + ... + b_nx_n \qquad \text{Eq(1)}$$

$$P = 1/(1 + e^{-(a+b1x1+b2x2+...+bnxn)}) \qquad \text{Eq(2)}$$

Where $x_1, ..., x_n$ is the predictor variables and 'P' is the proportion of food insecure household.

**Table 2. Socio-demographic characteristics of Urban Productive Safety Net Program beneficiary households in Addis Ababa, Ethiopia, June to July 2019.**

| Variable | Category | Frequency (*n*) | Percentage (%) |
|---|---|---|---|
| **Sex** | Female | 382 | 62.9 |
| | Male | 225 | 37.1 |
| **Age (years)** | <40 | 260 | 42.8 |
| | 40–60 | 268 | 44.2 |
| | >60 | 79 | 13 |
| **Marital status** | Unmarried | 62 | 10.2 |
| | Widowed | 107 | 17.6 |
| | Divorced | 121 | 19.9 |
| | Married | 317 | 52.2 |
| **Educational status** | No education | 154 | 25.4 |
| | Primary level | 259 | 42.7 |
| | Secondary level | 107 | 17.6 |
| | Above secondary | 87 | 14.3 |
| **Family size (persons)**[*] | 1–3 | 245 | 40.4 |
| | 4 or above | 362 | 59.6 |
| **Dependency ratio** | Low | 238 | 39.2 |
| | Middle | 286 | 47.1 |
| | High | 83 | 13.7 |

[*]Mean family size (persons) = 4.04±1.75

To find out the factors associated with food insecurity, data were analysed using binary logistic regression model at 95%CI. From the bivariate analysis, factors having a *p*-value of less than 0.25 were included in the multivariable analysis. By adjusting the confounders, from the multivariable logistic regression analysis, any variable at *p*-value <0.05 with 95% CI was declared a factor significantly associated with household food insecurity. Multicollinearity was checked among independent variables using standard error; a standard error of more than 2 indicated multicollinearity (maximum SE = 0.56). Model fitness was also checked using the Hosmer-Lemeshow test with *p*-value >0.05 (*p* = 0.858).

**Ethics approval and consent to participate.** Ethical clearance was obtained from the Ethical Review Board of Saint Paul's Hospital Millennium Medical College, Addis Ababa. Before the data collection started, an official request for permission to proceed was granted by the Addis Ababa City UPSNP administration office. Before each interview, data collectors explained the purpose of the study, gave assurance of confidentiality, addressed any other ethical issues and obtained informed verbal consent from each household respondent.

## Results

### Socio-demographic and economic characteristics of the study participants

Of 624 participants, 607 participated in this study (97.3% response rate). The mean age of household head and mean family size were 44.66±13.16 and 4.04±1.75, respectively. Nearly two-thirds 382 (62.9%) of households were headed by a female. One-fourth 154 (25.4%) of household heads were illiterate. Eighty-three (13.7%) households were found in the high dependency ratio group. Just more than half 317 (52.2%) of study participants were married (Table 2). Three hundred seventy-nine (62.4%) households' only source of income was from the UPSNP safety net. The mean household monthly income was 65.13±1.18 USD The

**Table 3. Socioeconomic characteristics of Urban Productive Safety Net Program beneficiary households in Addis Ababa, Ethiopia, June to July 2019.**

| Variable | Category | Frequency (*n*) | Percentage (%) |
|---|---|---|---|
| **Source of food** | Prepared at home | 567 | 93.4 |
| | Bought at restaurant | 31 | 5.1 |
| | Other | 9 | 1.5 |
| **Gender responsible for purchase of food** | Male | 85 | 14 |
| | Female | 522 | 86 |
| **Ownership of the house** | Private rental | 143 | 23.6 |
| | Government rental | 347 | 57.2 |
| | Privately owned house | 96 | 15.8 |
| | Other | 21 | 3.5 |
| **Sources of income** | From UPSNP only | 379 | 62.4 |
| | From daily labor and UPSNP | 83 | 13.7 |
| | From pension and UPSNP | 18 | 3 |
| | From self-employment and UPSNP | 127 | 20.9 |
| **Access to credit service** | No | 557 | 91.8 |
| | Yes | 50 | 8.2 |
| **Access to free health service** | No | 580 | 95.6 |
| | Yes | 27 | 4.4 |
| **Average monthly household income** | Low | 265 | 43.7 |
| | Medium | 154 | 25.4 |
| | High | 188 | 31 |
| **Proportion of monthly household income spent on food** | <75% | 176 | 29 |
| | ≥75% | 431 | 71 |
| **Household member had history of chronic medical problem***  | Yes | 143 | 23.6 |
| | No | 464 | 76.4 |

*History of chronic medical problems included one or more of diabetes, hypertension, HIV/AIDS, TB (tuberculosis bacillus), mental disorder or other chronic health problems.

proportion of household income spent on food and food-related items was more than 75% for 431 (71%) of households (Table 3).

**The occurrence of Household Food Insecurity Access Scale conditions.** The proportion of households experiencing worry about not having enough food was 528 (87%). The majority of the households 456 (75%) replied affirmatively to having been unable to eat their preferred food in the 4 weeks before the interview due to a lack of resources (Fig 1). The number of households that had eaten a limited variety of foods in the 4 weeks before the interview due to lack of resources was 462 (76.1%). Households that had eaten smaller amounts at a meal or fewer meals than preferred in the 4 weeks before the interview numbered 426 (70.2%) and 383 (63.1%), respectively. Moreover, the number of households with an affirmative response to the severe conditions of going to sleep hungry or going a whole day and night without food were 219 (36.1%) and 39 (6.4%), respectively (Table 4).

## The occurrence of household food insecurity domain

The nine occurrence questions were grouped into three domains according to similarities of their characteristics. the proportion of households falling into the anxiety domain was 87% (528), insufficient quality of food domain 76.3% (463) and insufficient quantity of food and physical consequences domain 72.5% (440) (Fig 2).

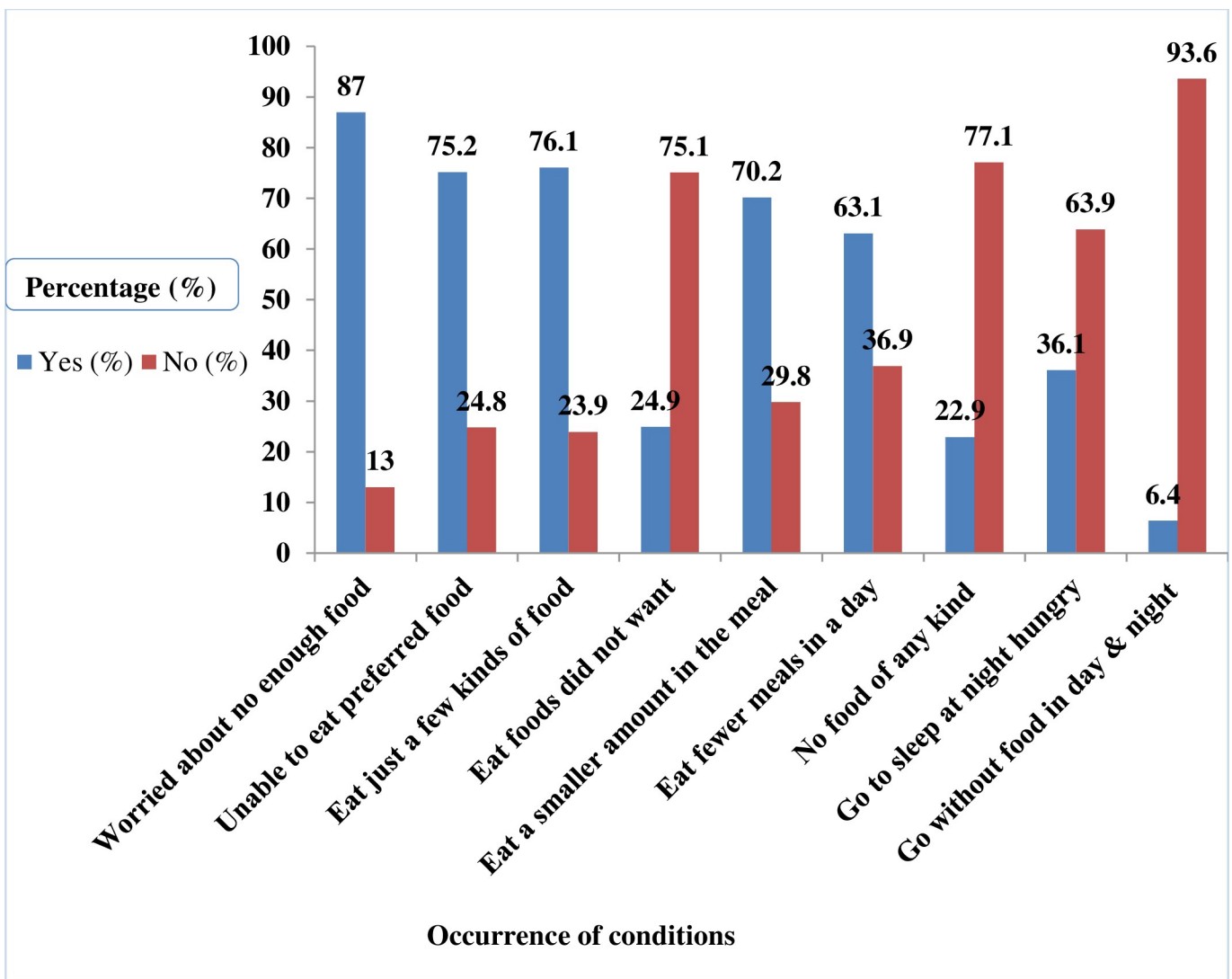

**Fig 1. Occurrence of Household Food Insecurity Access Scale conditions among Urban Productive Safety Net Program beneficiary households in Addis Ababa, June 2019.**

### Prevalence of household food insecurity

The overall prevalence of household food insecurity was 77.1% (468) with 95% CI: 73.8–80.7. Four hundred sixty-eight (77.1%) households had experienced some extent of food insecurity in the four weeks before the survey; mildly food insecure 4 (0.7%), moderately food insecure 129 (21.3%) and severely food insecure 335 (55.2%), while 139 (22.9%) households were food secure (Fig 3).

**Factors associated with household food insecurity.** Based on the result of multivariable analysis, households headed by an uneducated person was 2.56 times (AOR: 2.56; 95%CI: 1.08–6.07) more likely to be food insecure than households headed by individuals who had an education above secondary level, and the odds of food insecurity among households headed by a person with completed secondary-level education were 3.22 times greater (AOR: 3.22; 95% CI: 1.54–6.75) than among households headed by someone who had completed above a secondary level of education (Table 5).

**Table 4. Occurrence of HFIAS affirmative condition among UPSNP beneficiary households in Addis Ababa, Ethiopia, June to July 2019.**

| HFIAS affirmative questions | Affirmative response (yes) | | | |
|---|---|---|---|---|
| | Rarely | Sometimes | Often | Total |
| | *n*(%) | *n*(%) | *n*(%) | *n*(%) |
| Worried about not enough food | 86(14.2) | 114(18.8) | 328(54.0) | 528(87.0) |
| Unable to eat preferred food | 19(3.1) | 121(19.9) | 316(52.1) | 456(75.2) |
| Ate just a few kinds of food | 14(2.3) | 92(15.2) | 356(58.6) | 462(76.1) |
| Ate unwanted kinds of food | 51(8.4) | 82(13.5) | 18(3.0) | 151(24.9) |
| Ate smaller amount than desired at the meal | 20(3.3) | 177(29.2) | 229(37.7) | 426(70.2) |
| Ate fewer meals than desired in a day | 50(8.2) | 175(28.8) | 158(26.0) | 383(63.1) |
| Had no food of any kind for a day | 87(14.3) | 47(7.7) | 5(0.8) | 139(22.9) |
| Went to sleep at night hungry | 83(13.7) | 130(21.4) | 6(1.0) | 219(36.1) |
| Went without food for a day & night | 38(6.3) | 1(0.2) | 0(0.0) | 39(6.4) |

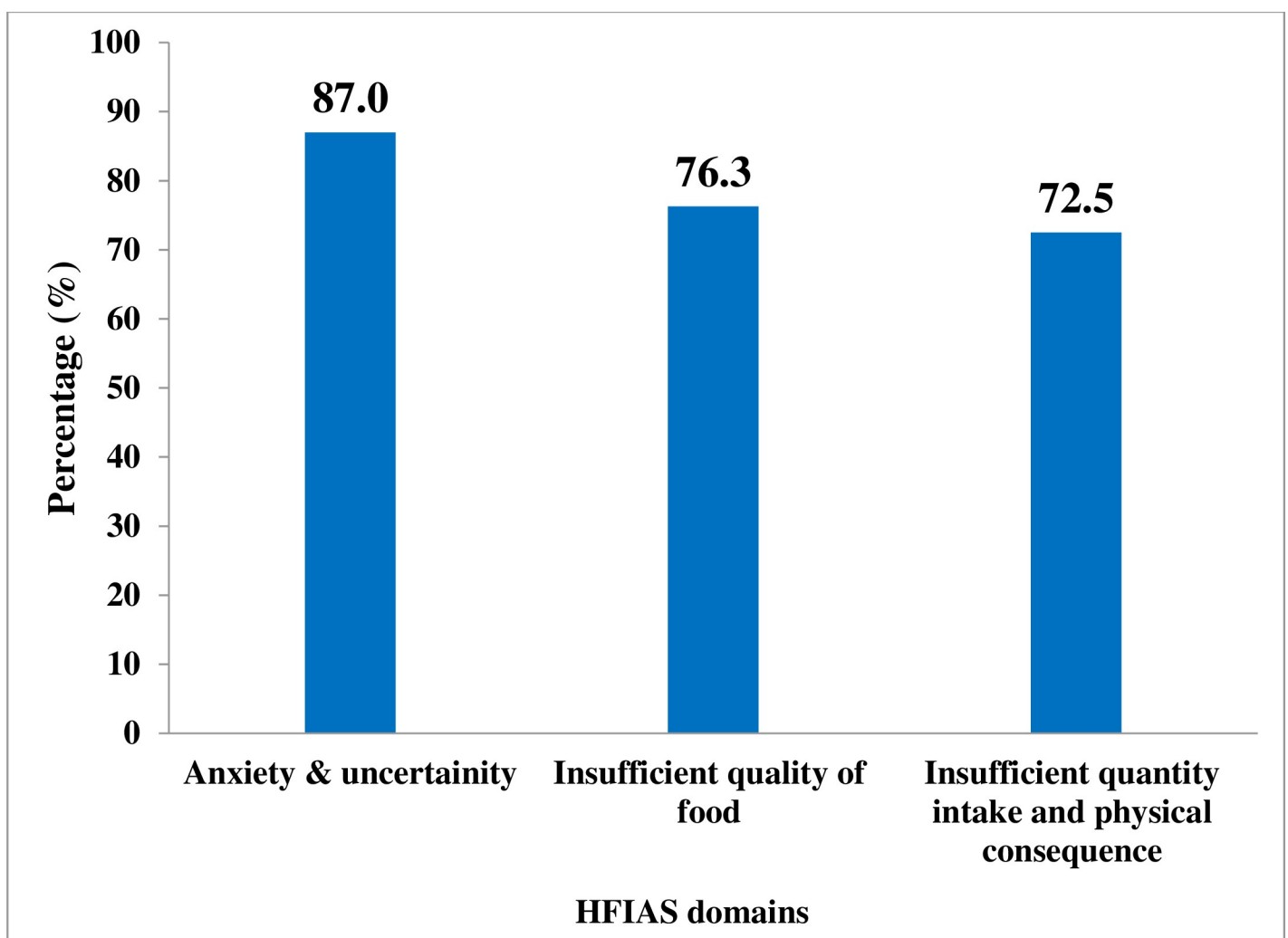

**Fig 2. Household Food Insecurity Access Scale domain distribution among Urban Productive Safety Net Program beneficiary households in Addis Ababa, June 2019.**

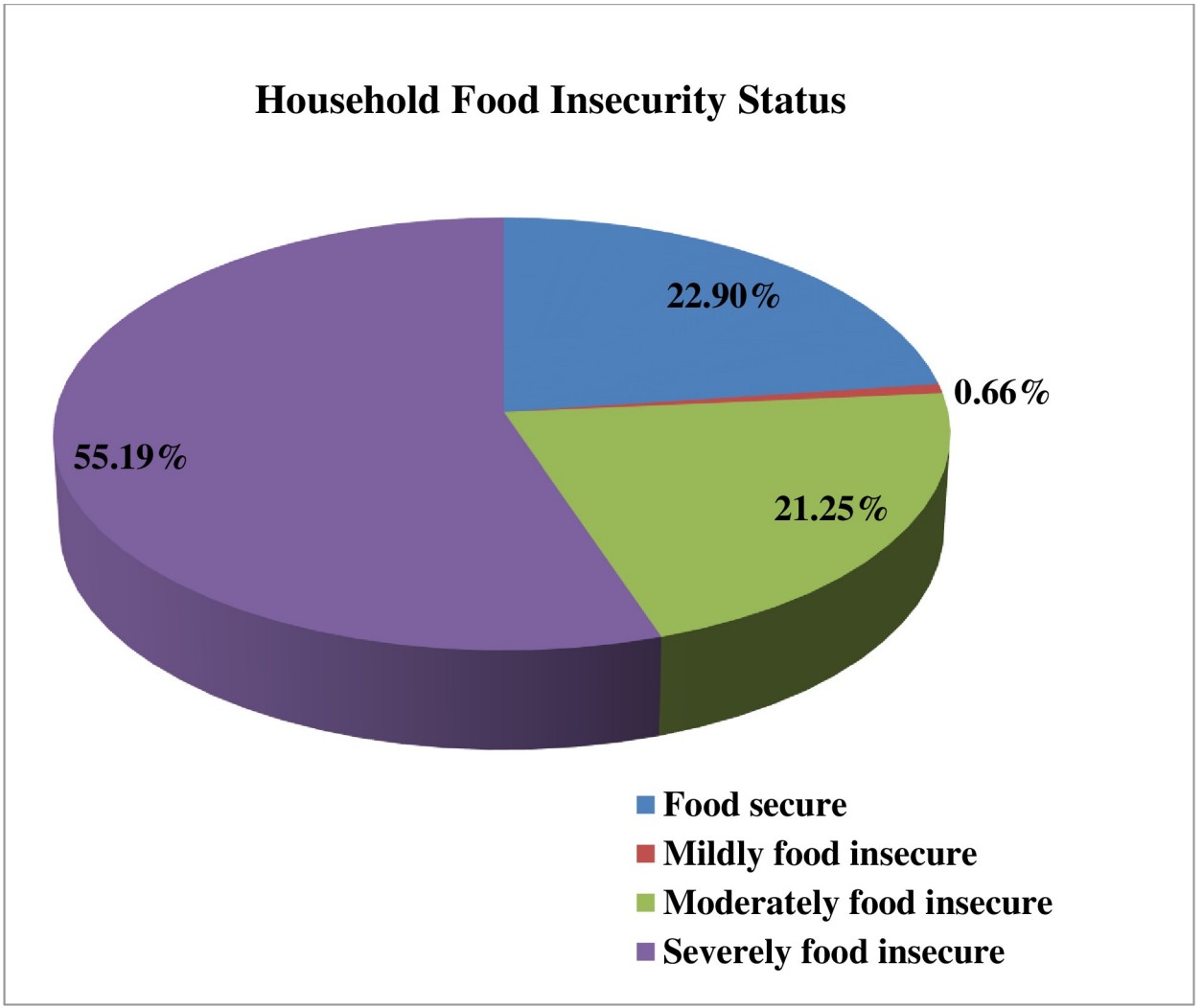

**Fig 3. Household food insecurity status among Urban Productive Safety Net Program beneficiary households in Addis Ababa, June 2019.**

Households with 4 or more family members were nearly twice (AOR: 1.87; 95%CI: 1.08–3.23) as likely to suffer from food insecurity as households having fewer than 4 family members. Likewise, households with a high dependency ratio were almost 4 times (AOR: 3.95; 95%CI: 1.31–11.90) as food insecure as households with a low dependency ratio. Households with no access to credit services were nearly three times [AOR: 2.85; 95%CI:1.25–6.49] more likely to develop food insecurity than households that had access to credit services (Table 5).

The odds of food insecurity among those with a low household income were nearly 5 times (AOR: 4.72; 95%CI: 2.32–9.60) higher than those with high household income; moreover, households with a medium income were 10 times (AOR: 9.78; 95%CI: 4.29–22.35) more likely to experience food insecurity than household with a high income (Table 5).

## Discussion

A HFIAS method was carried out using a cross-sectional study to assess the prevalence of household food insecurity among UPSNP beneficiary households in Addis Ababa;. We found that the prevalence of food insecurity was 77.1% and factors significantly associated with food

**Table 5. Factors significantly associated with household food insecurity from multivariable logistic regression analysis.**

| Variables* | AOR (95% CI) | p-value |
|---|---|---|
| **Educational status** | | |
| No education | 2.56(1.08–6.07) | 0.033 |
| Primary level | 3.22(1.54–6.75) | 0.002 |
| Secondary level | 1.00(0.48–2.09) | 0.988 |
| Above secondary level | 1 | |
| **Family size (persons)** | | |
| 1–3 | 1 | |
| 4 or above | 1.87(1.08–3.23) | 0.025 |
| **Dependency ratio** | | |
| Low | 1 | |
| Middle | 1.57(0.91–2.71) | 0.106 |
| High | 3.95(1.31–11.90) | 0.015 |
| **Access to credit service** | | |
| No | 2.85(1.25–6.49) | 0.013 |
| Yes | 1 | |
| **Household income** | | |
| Low | 4.72(2.32–9.60) | <0.001 |
| Middle | 9.78(4.29–22.35) | <0.001 |
| High | 1 | |

*Variables from the bivariate analysis that had P-value<0.25 were adjusted for multivariable analysis: age of household head; marital status of household head; sex of household head; educational status of household head; household family size; dependency ratio; history of medical problem; main sources of household food; gender of food purchaser; proportion of income spent on food; ownership/rental of house; sources of household income; average monthly household income; access to credit services and access to free health services.

insecurity were education level, dependency ratio, access to credit service and household income.

The prevalence of food insecurity (77.1%) found in this study was consistent with a previous study in Addis Ababa (75.0%) [30], East Badawacho District, South Ethiopia (75.8%) [31] and Sekela District, Western Ethiopia (74.1%) [32], but higher than found in several other studies in Addis Ababa (58.16%) [24], Wolaita Soda (37.6%) [33], West and East Gojjam (55.3%) [34]. The higher prevalence in our study might be attributable to our study being conducted among poor households that were UPSNP beneficiaries, whereas the households in other studies were from different parts of Ethiopia that were not safety net beneficiaries. In Addis Ababa, being poor is key to eligibility for the UPSNP safety net beneficiary program. The variations of these results indicate that area-specific food insecurity surveys are very important to visualize the differences in people's real situations.

Furthermore, the higher degree of household food insecurity found in this study relative to a national prevalence of food insecurity survey (35.0%) [35] might be due to our very small study area compared to the large coverage of the Ethiopian studies, which allows our findings to give a better insight about food insecurity status in the studied areas.

Our study revealed that 70.2% of households ate smaller amounts at meals than preferred, 63.1% of households also missed some meals during the studied period and 36.1% of the households also reported they went to bed hungry during the survey period. These results were higher than those of a study conducted in East Badawacho, South Ethiopia [31] that

found 62.3% and 10.7% of households were eating a smaller amount of food and going to sleep at night hungry, respectively. The increment of households experiencing food insecurity conditions might be due to householders' low purchasing power to access the available food and also the high inflation of food prices at the time of this survey, forcing householders to reduce the variety and amount of food they consumed, especially low-income urban households that spent a large proportion of their income to purchase food.

Our findings of the households falling in anxiety and uncertainties domain (87.0%), insufficient food quantity domain (76.3%), and insufficient food intake and its physical consequence domain (72.5%) were higher than the findings in Wolaita Sodo Town [33] that reported the percentage of households falling in these domains were 37.3%, 37.6% and 33.3%, respectively; and in West Abaya District 38.1%, 38.1%, and 34.5%, respectively [36]. These discrepancies may be attributed to the possibility that vulnerability to food insecurity among the urban poor is higher than among semi-urban and rural households. It can be also explained by the difference that semi-urban and rural households can access food from their garden production; in contrast, urban poor households are dependent on purchasing from the market food supply.

This study's findings that a household head's educational status of none or primary school completed as significantly associated with household food insecurity is consistent with other studies [12, 36, 37]. This might be because when the householder's level of education increases, there is access to better job opportunities and boosting of labour productivity that ultimately increases household income and provides an asset to safeguard the access to and utilization of food.

In this study, having a large number of household members increased the likelihood of food insecurity, a finding similar to that of studies conducted in Addis Ababa, South Africa, and Latin America [2, 24, 38, 39]. In contrast to our study, studies in Kenya and other rural parts of north Ethiopia found that households with a large family size were more like to be food secure as compared to those with a small family size [40, 41]. This discrepancy may be due to the fact that as the household size becomes larger, the number of economically active members may also increase; and the increased participation in income-generating opportunities leads to more food security than for households with lower family size.

Households with a high dependency ratio have a greater chance of being food insecure than those with a low dependency ratio. This might be because the dependent members are less likely to participate in productive work and their full dependence on others to purchase food and other commodities increases the burden on the household. Our findings were similar to those of several studies in Africa [33, 38, 40].

We also found that lack access to credit services was one of the factors associated factors of food insecurity as also reported by other studies in Ethiopia [24, 37]. This may be due to households having access to credit participating in diversified income-generating opportunities that are able to boost the financial power of the household. Access to credit services also helps households to cope with the food shortage situation by stabilizing the food purchasing powers of the household. We also found that households in low- and medium-income categories were more likely to experience food insecurity than those who had high household income, a finding consistent with previous studies conducted globally, including a study in Ethiopia [2, 12, 33, 38]. The low household income can induce food insecurity; as a vicious cycle, food insecurity also reduces household productivity.

## Limitation of the study and future research directions

The estimation of prevalence of household food insecurity relied on a one-month recall of occurrence; recall and social desirability bias could increase or decrease its reported

prevalence. However, to overcome such limitations, we used HFIAS, a validated, highly reliable and culturally sound tool that was able to minimize bias and reveal a relatively good estimate of the prevalence. An attempt also was made to minimize the possibility of bias by explaining the main objectives of the study and by applying appropriate data collection procedures and interview techniques through the use of trained data collectors and supervisors.

Furthermore, the absence of baseline data for food insecurity levels before the UPSNP began in Addis Ababa limits our ability to compare the levels of food insecurity in our study with those before the program began. In the future, the question of whether the UPSNP safety net program creates an attitude of dependency on aid will be useful to investigate in order to provide possible help designing mechanisms to end the need for beneficiaries to receive the aid for so long.

Future researchers are also encouraged to investigate the consequences of the UPSNP such as conflict between beneficiaries and non-beneficiaries, food price inflation, drought, capability of the program beneficiaries, and internal population movements and more. It is also recommended that a study be conducted on the cause and effect relationship between the UPSNP and poverty to ensure that urban food security is addressed at the country level over time.

## Implications of the study for practice and/or policy

The rural safety net program is an older project in Ethiopia, whereas the urban safety net program UPSNP is a new program scaled up from the rural experience and started as a pilot in 2017 in urban Ethiopia. One of the strengths of this study was that it investigated the food insecurity status of beneficiary households within the pilot project area in Addis Ababa. The findings will have practical application in helping to design a sustainable urban safety net program that improves the livelihoods of current beneficiaries and others in the future. To make a change on the level of food insecurity, this study shows that implementers should give due attention to variables such as household income levels, number of dependent members of beneficiary households, and household access to credit.

Furthermore, the evidence in this study may help programmers and policymakers to give priority to solving the challenges of the predictors for household food insecurity it revealed. Considering the prevalence of food insecurity and associated factors among UPSNP beneficiary households in Addis Ababa, the findings may help the city's UPSNP agency in collaboration with governmental and non-governmental organizations to further scale up programs that will help households help themselves in a sustainable manner. Further, the findings regarding beneficiaries of this urban safety net program may guide its expansion to other urban areas of Ethiopia, thereby reducing national food insecurity levels, and making progress toward the UN Sustainable Development Goal to end all forms of malnutrition.

## Conclusions

This study showed that three in four households were experiencing food insecurity and that an appropriate multi-dimensional approach including such interventions as improving educational status, creating employment opportunities or the opening of small business enterprises through credit service support may help to diversify income sources and to reduce the dependency ratio. It is recommended that the UPSNA do an impact evaluation of the current safety net program and consider increasing the amount of the per-person monthly payment, which in turn would increase the purchasing power of urban poor households. It is also highly recommended that the urban safety net program give special focus on the creation of sustainable income opportunities for users to improve their means of self-support.

## Supporting information

**S1 File. English version of the questionnaire for the survey of food insecurity status and determinants among Urban Productive Safety Net Program beneficiary households in Addis Ababa, Ethiopia.**
(DOCX)

**S2 File. Amharic version of the questionnaire for the survey of food insecurity status and determinants among Urban Productive Safety Net Program beneficiary households in Addis Ababa, Ethiopia.**
(DOCX)

**S1 Dataset. Data set for the survey of food insecurity status and determinants among Urban Productive Safety Net Program beneficiary households in Addis Ababa, Ethiopia.**
(XLSX)

## Acknowledgments

We acknowledge Addis Ababa City Urban Productive Safety Net Program Agency for giving us relevant information for this study. We also highly appreciate the study area's district administrators for their support by providing necessary information. We thank data collectors, supervisors and household members for their voluntary participation in the study. Last but not least, we also thank Lisa Penttila for language editing of the manuscript.

## Author Contributions

**Conceptualization:** Atimen Derso, Hailemichael Bizuneh, Awoke Keleb, Ayechew Ademas, Metadel Adane.

**Data curation:** Atimen Derso, Hailemichael Bizuneh, Awoke Keleb, Ayechew Ademas, Metadel Adane.

**Formal analysis:** Atimen Derso, Hailemichael Bizuneh, Awoke Keleb, Ayechew Ademas, Metadel Adane.

**Funding acquisition:** Atimen Derso, Hailemichael Bizuneh, Awoke Keleb.

**Investigation:** Atimen Derso, Hailemichael Bizuneh, Awoke Keleb, Ayechew Ademas, Metadel Adane.

**Methodology:** Atimen Derso, Hailemichael Bizuneh, Awoke Keleb, Ayechew Ademas, Metadel Adane.

**Project administration:** Atimen Derso, Hailemichael Bizuneh, Awoke Keleb, Ayechew Ademas, Metadel Adane.

**Resources:** Atimen Derso, Hailemichael Bizuneh, Awoke Keleb, Ayechew Ademas, Metadel Adane.

**Software:** Atimen Derso, Hailemichael Bizuneh, Awoke Keleb, Ayechew Ademas, Metadel Adane.

**Supervision:** Atimen Derso, Hailemichael Bizuneh, Awoke Keleb, Ayechew Ademas, Metadel Adane.

**Validation:** Atimen Derso, Hailemichael Bizuneh, Awoke Keleb, Ayechew Ademas, Metadel Adane.

**Visualization:** Atimen Derso, Hailemichael Bizuneh, Awoke Keleb, Ayechew Ademas, Metadel Adane.

**Writing – original draft:** Atimen Derso, Hailemichael Bizuneh, Awoke Keleb, Ayechew Ademas, Metadel Adane.

**Writing – review & editing:** Hailemichael Bizuneh, Awoke Keleb, Ayechew Ademas, Metadel Adane.

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
