## [Decision Letter · Decision Letter 0]

16 Nov 2020

PONE-D-20-30124

Food Insecurity Status and Determinants among Urban Productive Safety Net Program Beneficiary Households in Ethiopia

PLOS ONE

Dear Dr. Adane (PhD),

Thank you for submitting your manuscript to PLOS ONE. After careful consideration, we feel that it has merit but does not fully meet PLOS ONE’s publication criteria as it currently stands. Therefore, we invite you to submit a revised version of the manuscript that addresses the points raised during the review process.

We look forward to receiving your revised manuscript.

Kind regards,

Yacob Zereyesus, Ph.D.

Academic Editor

PLOS ONE

Journal Requirements:

2. Please include additional information regarding the survey or questionnaire used in the study and ensure that you have provided sufficient details that others could replicate the analyses. For instance, if you developed a questionnaire as part of this study and it is not under a copyright more restrictive than CC-BY, please include a copy, in both the original language and English, as Supporting Information, or include a citation if it has been published previously.

3. In statistical analyses, please clarify whether you corrected for multiple comparisons.

4. In statistical analyses, did you consider use of survey weights to account for the sampling design? If not, please provide your rationale.

5.We suggest you thoroughly copyedit your manuscript for language usage, spelling, and grammar. If you do not know anyone who can help you do this, you may wish to consider employing a professional scientific editing service.  

Reviewers' comments:

Reviewer's Responses to Questions

**Comments to the Author**

1. Is the manuscript technically sound, and do the data support the conclusions?

Reviewer #1: Partly

Reviewer #2: No

Reviewer #3: Yes

2. Has the statistical analysis been performed appropriately and rigorously? 

Reviewer #1: Yes

Reviewer #2: No

Reviewer #3: Yes

3. Have the authors made all data underlying the findings in their manuscript fully available?

Reviewer #1: Yes

Reviewer #2: No

Reviewer #3: No

4. Is the manuscript presented in an intelligible fashion and written in standard English?

Reviewer #1: Yes

Reviewer #2: Yes

Reviewer #3: Yes

5. Review Comments to the Author

Reviewer #1: The paper presents useful information on food security. A number of comments address the need to more clearly source and date data presented, especially when using those data to support claims that the food insecurity situation is worsening. The absence of baseline data for food insecurity levels in 2017, when the program began, limits the ability to directly compare the levels of food security reported in the survey results with those before the program began. As such the findings report one month’s helpful. A table that clearly shows the three levels of food insecurity and the number used for the dichotomous variable used in the analysis would be useful. It is possible that some of this is in Table 4 which I could not access.

Dr. Cheryl Christensen

Reviewer #2: The study addresses a very important topic i.e. understanding food insecurity in urban Ethiopia using a household survey data. It presents an analysis of the food insecurity status or urban households in Ethiopia and examines its determinants. The study finds that the key determinants of food insecurity include education, family size, access to credit and income.

But the study does not address significant problems in sample selection, estimation, and presentation of results. The following comments could provide some recommendations, if followed could improve the chances of publication in the future.

- Study notes that a sample of 624 households were selected from the 78,543 households from the Urban Productive Safety Net Program (PSNP) Beneficiary Households using a “single proportion formula”. It is not clear why this sample size is chosen. Please provide a solid framework to justify your sample choice. It would also be good to provide a summary statistic of the sample households in comparison to the entire data set since the PSNP data is already available.

- It would be good to state from the outset what the key objectives of the study are and the value added of the study. The study goes on describing food insecurity in the study area without providing the key objectives and contribution of this particular study in relation to existing literature. Identifying the contribution of this particular study would help readers follow the logic in the paper.

o Is it that we don’t have good evidence in urban areas?

o Is it the food insecurity measure used that makes this study unique and contributing to existing studies in the literature?

- A concise and clear description of the food insecurity measure used could improve the study since that could be one of the key contributions. You mention that you use a “tool that has been developed by Food and Nutrition Technical Assistance (FANTA). Here, provide why this measure is preferred to other measures of food insecurity. The PSNP itself may already have other indicators that do not require collecting additional data. How does this measure compare to this measure?

- The section “Data Quality Assurance” and “Data Management and Statistical Analysis” could be replaced by a methods section that clearly specifics the following

o Data: Sample selection, choice of indicators (including food insecurity measure), description of study area (urban Addis Ababa for context)

o Estimation: method of estimation (you indicated logistic), justify choice of method, benchmark model (equation), definition of variables (already in previous section)

Though the study addresses an important topic, it requires significant revision to be considered for publication. If the author addresses the issues indicated above and provides greater clarity in the write up, the chances of publication will be improved either in this journal or elsewhere.

Reviewer #3: Comments to the authors

This study is very important in terms of developing policies regarding tackling household food insecurity from developing countries perspective. However, this study has been nicely presented, need some focus on the following points,

1. Introduction is too long to read. Some information are surplus like adult and child malnutrition etc. Please provide precious information to make the introduction short.

2. This study is lacking the findings of previous studies to assess the appropriate gap. Please add the proper theoretical framework.

3. I don’t think the section “operational definition” section is adding much in this study. Better to add the study variable list.

4. The “Methodology” section is too long. Data collection tools and procedure and data quality assurance could be readjusted.

5. “Socio-demographic Characteristics of the Study Participants” and “Socioeconomic Characteristics of Study Participants” do not add much significant information in this study. So, need to reorganize in to one para and highlight only key findings.

6. “The occurrence of household food insecurity domain” this section is unclear. How it measured.

7. Why did not you compare the results between UPSNP safety net beneficiary and other available samples.

8. There are numerous studies in many developing countries regarding this issue. What is the difference between current study and previous study except study population?

6. PLOS authors have the option to publish the peer review history of their article (what does this mean?). If published, this will include your full peer review and any attached files.

Reviewer #1: **Yes: **Dr. Cheryl Christensen

Reviewer #2: No

Reviewer #3: No

---

## [Author Response · Author response to Decision Letter 0]

8 Mar 2021

Date: March 08, 2021

Manuscript ID: PONE-D-20-30124

Food Insecurity Status and Determinants among Urban Productive Safety Net Program Beneficiary Households in Addis Ababa, Ethiopia

Corresponding authors: Metadel Adane (PhD)

Dear Dr. Yacob Zereyesus, Ph.D., Academic Editor

Dear Reviewers, 

PLOS ONE

Thank you all for your letter dated 16 Jan 2020 with a decision of revision required. We were pleased to know that our manuscript was considered potentially acceptable for publication in PLoS ONE, subject to adequate revision as requested by the reviewers, academic editor and the journal. Based on the instructions provided in your letter, we uploaded the file of the rebuttal letter; the marked up copy of the revised manuscript highlighting the changes made in the original submitted version and the clean copy of the revised manuscript. 

We have revised the manuscript by modifying the abstract, introduction, methods, results, discussion and other sections, based on the comments made by the reviewers and using the journal guidelines. Accordingly, we have marked in red color all the changes made during the revision process. Appended to this letter is our point-by-point response (rebuttal letter) to the comments made by the reviewers. 

We agree with almost all the comments/questions raised by the reviewers and provided justification for disagreeing with some of them. We would like to take this opportunity to express our thanks to the reviewers for their valuable comments and to thank you for allowing us to resubmit a revision of the manuscript. 

I hope that the revised manuscript is accepted for publication in PLoS ONE. 

Sincerely yours,

Metadel Adane (PhD) 

Rebuttal letter

Response to the Journal Requirements

Response: We thank you for your key comments and we revised the manuscript accordingly POLS ONE manuscript preparation templates including file naming (Please see the revised version). 

2. Please include additional information regarding the survey or questionnaire used in the study and ensure that you have provided sufficient details that others could replicate the analyses. For

instance, if you developed a questionnaire as part of this study and it is not under a copyright more restrictive than CC-BY, please include a copy, in both the original language and English, as Supporting Information., or include a citation if it has been published previously.

Response: We provided the survey tool or questionnaire as supportive information in both English and Amharic (original language) version labeled as S1 and S2, respectively (Please see on the revised version). 

3. In statistical analyses, please clarify whether you corrected for multiple comparisons.

Response: Thank you for this key comment. We used adjusted analysis during multivariable logistic regression analysis and please see the revised version in page 14 from lines 321 to 326. 

4. In statistical analyses, did you consider use of survey weights to account for the sampling design? If not, please provide your rationale.

Response: Thank you again for your valuable comment. Our study is purely cross-sectional survey which is not a type of cluster study. First proportional to size allocation of samples were performed to account sample size allocation in a representative manner. Thus, using survey weights is not used in our study due to the study design and sampling methods type. Cluster weights are common for a study that has a cluster type. 

5. We suggest you thoroughly copyedit your manuscript for language usage, spelling, and grammar. If you do not know anyone who can help you do this, you may wish to consider employing a professional scientific editing service.

Response: Many thanks for this comment and we addressed the issue of language usage, spelling and grammar carefully by getting support from the experience language editor as you can see in our acknowledgment section in page 22 from lines 553 to 534. 

6. Please provide the name of the colleague or the details of the professional service that edited your manuscript

Response: Lisa Penttila, who is from Canada is edited our manuscript. 

7. Provide a copy of your manuscript showing your changes by either highlighting

them or using track changes (uploaded as a *supporting information*

file)

Response: We revised the manuscript by highlighting the change and labeled as * ‘manuscript showing your changes 

8. Provide a clean copy of the edited manuscript (uploaded as the new *manuscript* file)

Response: We prepared clean copy of the revised manuscript file, labeled as the new*manuscript* file

Response to reviewers

Reviewer # 1

Comment on the paper

Reviewer # 1-The paper presents useful information on food security.

Response: Thank you for the positive remark on our paper and we really appreciate your scientific judgment. Regarding to other comments, we improved the manuscripts by considering all your concerns. We found that all comments are very useful and thank you again. 

Reviewer # 1-A number of comments address the need to more clearly source and date data presented, especially when using those data to support claims that the food insecurity situation is worsening. The absence of baseline data for food insecurity levels in 2017, when the program began, limits the ability to directly compare the levels of food security reported in the survey results with those before the program began. As such the findings report one month’s helpful.

Response: Thank you very much for recognizing our study importance and you clearly pointed out the challenges of our study was lack of baseline data in 2017, where one month’s of our study is still very useful (Please see the revised version in the introduction section). 

 Reviewer # 1- A table that clearly shows the three levels of food insecurity and the number used for the dichotomous variable used in the analysis would be useful. It is possible that some of this is in Table 4 which I could not access. 

Response: We updated Table 4 to show the level of food insecurity and dichotomous variable (Please see the revised version table 4).

Reviewer# 2

Comment on the Paper

Reviewer #2: The study addresses a very important topic i.e. understanding food insecurity in urban Ethiopia using a household survey data. It presents an analysis of the food insecurity status or urban households in Ethiopia and examines its determinants. The study finds that the key determinants of food insecurity include education, family size, access to credit and income.

But the study does not address significant problems in sample selection, estimation, and presentation of results.

Response: Thank you for recognizing of our study addresses important issues about the urban food insecurity. We appreciate your positive reflection in our study. We accepted all your concerns and have addressed them accordingly. We found that the paper substantially improved because of your pertinent comments and thank you very much for your scientific input (please see the revised version to see all the updates regarding sample selection, estimation, presentation of results and other concerns). 

Question 1: Study notes that a sample of 624 households were selected from the 78,543 households from the Urban Productive Safety Net Program (PSNP) Beneficiary Households using a “single proportion formula”. It is not clear why this sample size is chosen. Please provide a solid framework to justify your sample choice. 

Response: We explained the sample size calculation process as follows. 

To estimate sample size, we considered the assumptions of sample estimator, previous prevalence (proportion of food insecurity), sample variability and limit of uncertainty (95% confidence interval). Here we assumed the UPSNP beneficiary population is approximately normally distribute then standard error of population value ‘p’ is 〖SE〗_p=√( P(1-P)/n) , 

then P(population value) = p(sample value) � z SEP

Zα/2 * SEP = variability of population value (W), 

 W = Zα/2 *SEP = z √( P(1-P)/n) 

Finally sample size was determined using a single proportion formula [n= (〖[ (z〗^(2 ) α⁄2)*(p(1-p)])/w^2 ] with the assumption of the prevalence (p) of household food insecurity was 58.16%, taken from a previous study in Addis Ababa, Ethiopia [22], 95% confidence interval (CI) (Za/2 = 1.96) and (w) marginal error (5%). A 1.5 design effect was also considered, and then 10% contingency for non-response rate included. Finally, we obtained a final sample size of 624 households (Please see the revised version in sample size determination in page 8 to 9).

Question 2: It would also be good to provide a summary statistic of the sample households in comparison to the entire data set since the PSNP data is already available.

Response: thank you, we introduced the sample summary statistics in the first paragraph of result section only for our sampled households because the detailed entire data set of PSNP beneficiary households in city is not available at central level rather only absolute number of beneficiary households are there at woreda level (please see the revised version). As reviewer #1 also noted and understood the situation, the absence of baseline data for food insecurity levels during the urban safety net program began in Addis Ababa limits us to compare the levels of food insecurity reported our study with those before the program began, which compromise to evaluate the change of food insecurity status due to the existing safety net program contribution (We noted this as one of the study the limitation in page 20 from lines 499to 503). 

Question 3: It would be good to state from the outset what the key objectives of the study are and the value added of the study. The study goes on describing food insecurity in the study area without providing the key objectives and contribution of this particular study in relation to existing literature. Identifying the contribution of this particular study would help readers follow the logic in the paper.

-Is it that we don’t have good evidence in urban areas?

-Is it the food insecurity measure used that makes this study unique and contributing to existing studies in the literature?

- A concise and clear description of the food insecurity measure used could improve the study since that could be one of the key contributions. 

Response: Dear reviewer, we thank you for these key comments and we agreed with all your ideas and the manuscript is already updated accordingly and please see revised version in the last two paragraphs of introduction section in page 6.

Question 4: You mention that you use a “tool that has been developed by Food and Nutrition Technical Assistance (FANTA). Here, provide why this measure is preferred to other measures of food insecurity. The PSNP itself may already have other indicators that do not require collecting additional data. How does this measure compare to this measure?

Response: You point is reasonable. Thank you. We made the revision according to your comment with the justification of why FANTA tool was used and please see the revised version in the outcome variable measurement in page 9 and 10. 

Question 5: The section “Data Quality Assurance” and “Data Management and Statistical Analysis” could be replaced by a methods section that clearly specifics the following

- Data: Sample selection, choice of indicators (including food insecurity measure), description of study area (urban Addis Ababa for context)

- Estimation: method of estimation (you indicated logistic), justify choice of method, benchmark model (equation), definition of variables (already in previous section)

Response: We updated these sections as per your comment. Please see the revised version in page 13 to 14. Thank you.

Reviewer #3

Reviewer #3: Comments to the authors

This study is very important in terms of developing policies regarding tackling household food insecurity from developing countries perspective. However, this study has been nicely presented, need some focus on the following points,

Response: We appreciate your positive reflection about the importance of our study and thank you for that. All your comments are addressed herewith. 

Question 1. Introduction is too long to read. Some information are surplus like adult and child malnutrition etc. Please provide precious information to make the introduction short.

Response: We agree with your comment. However, as you know food insecurity mainly affect children, which exposed for malnutrition for under-five children. To show the magnitude of the problem, we give emphasis about children. Furthermore, we also give background information about adult since food insecurity in urban areas also the main concern for adults. We tried to minimize the introduction, but still we feel that it is too much despite this does not have a problem for our study (See the revised version of the introduction). 

Question 2. This study is lacking the findings of previous studies to assess the appropriate gap. Please add the proper theoretical framework.

Response: We updated the introduction by including previous studies similar with our objectives to make strong the theoretical framework (See the updated version of the introduction)

Question 3. I don’t think the section “operational definition” section is adding much in this study. Better to add the study variable list.

Response: Operational definitions are very important to give information about measurement of variables (both outcome and independent variables) and feel that keeping it very nice for readers. Most of the key variables are listed within the operational definitions, we did not include some of the variables like socio-demographic since it is not as such complex to measure. 

Question 4. The “Methodology” section is too long. Data collection tools and procedure and data quality assurance could be readjusted.

Response: We merged as suggested. 

Question 5. “Socio-demographic Characteristics of the Study Participants” and “Socioeconomic Characteristics of Study Participants” do not add much significant information in this study. So, need to reorganize in to one para and highlight only key findings.

Response: We accept your comment, to make key hints for the readers, we write one paragraph about socio-demographic and economic characteristics of the study participants as you suggested (See in page 14 and 15). 

Question 6. “The occurrence of household food insecurity domain” this section is unclear. How it measured.

Response: To make it clear, we elaborate how the occurrence of household food insecurity domain was measured in outcome variable measurement section in page 9 to 11. Hoping that now it. Thank you very much. 

Question 7. Why did not you compare the results between UPSNP safety net beneficiary and other available samples? 

Response: We believe that this question is valid and we appreciate your observation. However, the absence of baseline data for food insecurity levels during the urban safety net program began in Addis Ababa limits us to compare the levels of food insecurity found our study with those before the program began, which compromise to evaluate the change of food insecurity status due to the existing safety net program contribution (We noted this as one of the study the limitation in page 20).

Question 8. There are numerous studies in many developing countries regarding this issue. What is the difference between current study and previous study except study population? 

Response: Dear reviewer, although there are many studies which has been conducted so far, ours is different regarding setting and its objectives that addressed, which is helpful in local context in Addis Ababa. The main objective of our study was to assess the level of food insecurity that could be reduced by transfer of cash to households at urban level. Urban poor’s are dependent on purchasing of food and related items so that transfer of cash at urban level has been designed to improve household income and ultimately improve level food security to the long term and response of emergency. So assessing status of food insecurity in pilot project area of UPSNP generated vital information that will helps the implementers to as an input evaluate the progress and give spotlight to readjust in sustainable manner and scale up the program, which all makes our study different from the other study regarding food insecurity. We explained this under the sub-themes of Implications of the study for practice and/or policy (Please see the revised version in page 21 to 22). 

We would like to thank the reviewers and editors for evaluating our manuscript. We have tried to address all the concerns in a proper way and believe that our paper has been improved considerably. We would be happy to make further corrections if necessary and look forward to hearing from you all soon. 

I hope that the revised manuscript is accepted for publication in PLoS ONE. 

Sincerely yours,

Metadel Adane (PhD in Water and Public Health)

---

## [Decision Letter · Decision Letter 1]

12 Aug 2021

Food Insecurity Status and Determinants among Urban Productive Safety Net Program Beneficiary Households in Addis Ababa, Ethiopia

PONE-D-20-30124R1

Dear Dr. Adane,

We’re pleased to inform you that your manuscript has been judged scientifically suitable for publication and will be formally accepted for publication once it meets all outstanding technical requirements.

Kind regards,

Abid Hussain

Academic Editor

PLOS ONE

Additional Editor Comments (optional):

Reviewers' comments:

Reviewer's Responses to Questions

**Comments to the Author**

1. If the authors have adequately addressed your comments raised in a previous round of review and you feel that this manuscript is now acceptable for publication, you may indicate that here to bypass the “Comments to the Author” section, enter your conflict of interest statement in the “Confidential to Editor” section, and submit your "Accept" recommendation.

Reviewer #1: All comments have been addressed

2. Is the manuscript technically sound, and do the data support the conclusions?

Reviewer #1: Yes

3. Has the statistical analysis been performed appropriately and rigorously? 

Reviewer #1: Yes

4. Have the authors made all data underlying the findings in their manuscript fully available?

Reviewer #1: Yes

5. Is the manuscript presented in an intelligible fashion and written in standard English?

Reviewer #1: Yes

6. Review Comments to the Author

Reviewer #1: (No Response)

7. PLOS authors have the option to publish the peer review history of their article (what does this mean?). If published, this will include your full peer review and any attached files.

Reviewer #1: **Yes: **Cheryl Christensen

---

## [Editor Report · Acceptance letter]

17 Sep 2021

PONE-D-20-30124R1 

Food Insecurity Status and Determinants among Urban Productive Safety Net Program Beneficiary Households in Addis Ababa, Ethiopia 

Dear Dr. Adane:

I'm pleased to inform you that your manuscript has been deemed suitable for publication in PLOS ONE. Congratulations! Your manuscript is now with our production department. 

Kind regards, 

on behalf of

Dr. Abid Hussain 

Academic Editor

PLOS ONE